# Phylogenetic and Expression Analysis of Fos Transcription Factors in Zebrafish

**DOI:** 10.3390/ijms231710098

**Published:** 2022-09-03

**Authors:** Khadizatul Kubra, Gurveer K. Gaddu, Clifford Liongue, Somayyeh Heidary, Alister C. Ward, Amardeep S. Dhillon, Faiza Basheer

**Affiliations:** 1School of Medicine, Deakin University, Geelong, VIC 3216, Australia; 2Institute of Mental and Physical Health and Clinical Translation, Deakin University, Geelong, VIC 3216, Australia; 3Olivia Newton-John Cancer Research Institute, Melbourne, VIC 3084, Australia; 4School of Cancer Medicine, LaTrobe University, Melbourne, VIC 3086, Australia

**Keywords:** Fos, activator protein-1, transcription factor, embryogenesis, zebrafish, development, ortholog

## Abstract

Members of the FOS protein family regulate gene expression responses to a multitude of extracellular signals and are dysregulated in several pathological states. Whilst mouse genetic models have provided key insights into the tissue-specific functions of these proteins in vivo, little is known about their roles during early vertebrate embryonic development. This study examined the potential of using zebrafish as a model for such studies and, more broadly, for investigating the mechanisms regulating the functions of Fos proteins in vivo. Through phylogenetic and sequence analysis, we identified six zebrafish *FOS* orthologues, *fosaa, fosab, fosb, fosl1a, fosl1b,* and *fosl2*, which show high conservation in key regulatory domains and post-translational modification sites compared to their equivalent human proteins. During embryogenesis, zebrafish *fos* genes exhibit both overlapping and distinct spatiotemporal patterns of expression in specific cell types and tissues. Most *fos* genes are also expressed in a variety of adult zebrafish tissues. As in humans, we also found that expression of zebrafish *FOS* orthologs is induced by oncogenic BRAF-ERK signalling in zebrafish melanomas. These findings suggest that zebrafish represent an alternate model to mice for investigating the regulation and functions of Fos proteins in vertebrate embryonic and adult tissues, and cancer.

## 1. Introduction

The mammalian FOS proteins, c-FOS, FOSB, FOSL1, and FOSL2, belong to the activator protein-1 (AP-1) family of transcription factors that contain evolutionarily conserved basic leucine zipper (bZIP) domains [1]. FOS proteins regulate transcription by forming heterodimers with other AP-1 proteins, particularly members of the JUN, ATF, and MAF families [2,3,4,5]. AP-1 complexes regulate the expression of genes important for cell proliferation, differentiation, and apoptosis in response to a plethora of extracellular signals, including growth factors, cytokines, and hormones [3,5,6,7,8,9,10,11]. In addition to possessing a conserved basic leucine zipper (bZIP) domain mediating DNA binding and dimerisation, FOS isoforms have different N-terminal and C-terminal regions, providing a basis for their differential regulation and functional activities [1]. c-FOS and FOSB have N- and C-terminal transactivation motifs termed the N-TA, C-TM, and TBD that are absent in FOSL1 and FOSL2. Additional C-terminal motifs HOB1 and HOB2, which stabilise and facilitate assembly of the pre-initiation complex, are present solely in c-FOS [1,12,13]. In contrast, FOSB has a unique proline-rich functional module, PRM also within the C-terminal region [14]. Another critical regulatory region present in all FOS proteins is the C-terminal destabilising element (C-DEST). Phosphorylation of this domain is induced by ERK MAPK signalling and leads to protein stabilisation [15,16,17].

FOS family members are implicated with the pathogenesis of various diseases [18]. Aberrant expression and activation of FOS proteins is driven by dysregulation of key cancer-associated signalling pathways, most notably the ERK MAPK pathway [1]. The pro-tumorigenic actions of FOS family members have been examined in a variety of in vitro and in vivo models, which have identified roles for FOS proteins in oncogenic transformation [19] and cancer progression [20,21,22,23]. Early studies showed that rodent fibroblasts undergo transformation upon expression of c-FOS but not FOSL1 and FOSL2, which is attributed to differences in the N- and C-terminal regions of these proteins. Notably, c-FOS and FOSB harbour transactivation domains that are absent in FOSL1 and FOSL2 [1,12]. Consequently, AP-1 dimers containing c-FOS and FOSB show stronger transcription activation potential than those containing FOSL1 and FOSL2 [24,25].

Genetic knockout and transgenic mouse models have shown that individual Fos proteins have specific in vivo functions [26,27]. *c-Fos* deficiency perturbs normal development of bone, cartilage, and the haematopoietic system [28], whereas its transgenic expression in the bone induces the formation of osteosarcoma [29]. *Fosb* deficiency impairs brain development, leading to nurturing defects in the adult female mouse, while transgenic overexpression of Fosb2 in the thymus disrupts T-cell differentiation [30]. Deficiency of either *Fosl1* and *Fosl2* in mice caused embryonic lethality, while their overexpression resulted in abnormalities in bone and ocular development [3,27]. *Fosl1* deletion also led to placental defects associated with the aberrant differentiation of trophoblasts [31,32].

Despite providing important insights into the in vivo functions of specific FOS proteins in normal and disease states, the use of mouse models to investigate the mechanisms regulating the activities of individual FOS isoforms in vivo is challenging. In addition, mouse models have provided limited insight into the functions of FOS isoforms during early vertebrate development. The zebrafish has emerged as a robust in vivo model for early developmental studies, as its embryos are transparent and develop externally, facilitating easy observation of key processes [33]. The zebrafish genome has been sequenced and about 70% of human genes have at least one ortholog in zebrafish. In addition, their rapid development, high fecundity, low maintenance costs, and amenability to genetic manipulation, including tissue-specific targeted genome editing and transgenesis, makes them well suited for investigating molecular mechanisms in vivo. Through phylogenetic and sequence analysis, we identified six zebrafish *FOS* homologues, *fosaa, fosab, fosb, fosl1a, fosl1b,* and *fosl2,* which encoded proteins showing high sequence conservation in key regulatory domains and post-translational modification sites compared to the equivalent human proteins. Spatiotemporal expression analysis revealed both overlapping and distinct patterns of *fos* gene in specific cell types and tissues during early embryonic development. Our data thus suggest that the zebrafish represents a valuable vertebrate model for investigating Fos protein regulation in vivo and defining their functions during embryogenesis.

## 2. Results

### 2.1. Identification of Zebrafish Fos Proteins

The human *FOS* gene family consists of *FOS*, *FOSB*, *FOSL1,* and *FOSL2* [1]. Analysis of genomic databases identified six zebrafish *FOS* genes, which synteny analysis indicated were single *fosb* and *fosl2* orthologues, but duplicated *fosaa/fosab* and *fosl1a/fosl1b* paralogues (Figure 1). Phylogenetic analysis of the encoded proteins showed they formed a distinct clade from another bZIP transcription factor (JDP2) outgroup (Figure 2) [34,35]. Within the FOS clades, there was a clear sub-clade for c-FOS, FOSB, and FOSL2 proteins, with the c-FOS paralogue Fosaa being more divergent than Fosab. In contrast, while Fosl1a grouped with its mammalian counterparts, Fosl1b sat as an outlier to all subclades. In addition, clades of c-FOS and FOSB as well as FOSL1 and FOSL2 proteins were grouped under two separate clades, indicating that these pairs of proteins share a common ancestral origin.

### 2.2. Major Functional Domains and Post-Translationally Modified Residues in Human FOS Proteins Are Conserved in Their Zebrafish Counterparts

Sequence alignment of full-length human, mouse, and zebrafish FOS proteins revealed that, in addition to the bZIP domain, other major N- and C-terminal functional domains of human FOS proteins [1] were present in their corresponding zebrafish proteins (Figure 3 and Appendix A). Protein BLAST analysis showed that human c-FOS was 48% identical to Fosaa and 55% to Fosab. For c-FOS, regions of significant homology were present throughout the protein, particularly in the bZIP motif, and to a lesser extent in the C- and N-terminal transactivation domains, N-TA, HOB1, HOB2, C-TM, and TBD (Figure 3A and Appendix A). Notably, previously characterised regulatory phosphorylation sites in mammalian FOS proteins are highly conserved in zebrafish Fosab but not in Fosaa, where only the C-terminal phosphorylation sites were conserved. The region of highest conservation in the c-FOS paralogs was the bZIP domain, while the C-DEST region of Fosab was more conserved to c-FOS than that of Fosaa (Appendix A).

The overall sequence identity of zebrafish FosB and human FOSB was 65%, with the bZIP domain being 93% identical. Amongst FOS proteins, FosB showed the highest overall sequence and bZIP domain conservation. Zebrafish FosB showed high conservation of the bZIP motif with mammalian FOSB and moderate conservation of the C- and N-terminal transactivation domains (N-TA, PRM, C-TM, and TBM) (Figure 3B and Appendix A). Key regulatory phosphorylation sites in human FOSB were also conserved in its zebrafish ortholog. However, the C-DEST region in zebrafish FosB is least conserved amongst FOS proteins.

In contrast to c-FOS and FOSB, human FOSL1 and FOSL2 lack potent transactivation domains [1,12]. The overall sequence identities of Fosl1a and Fosl1b compared to human FOSL1 are 49% and 40%, respectively, with their bZIP domain identities being 78% and 59%, respectively (Figure 3C). Finally, zebrafish Fosl2 showed 65% overall identity and 85% bZIP domain identity compared to human FOSL2 (Figure 3D). Of the two zebrafish FOSL1 paralogs, Fosl1b showed weaker sequence conservation compared to Fosl1a, with Fosl1b lacking the C-DEST domain and C-terminal phosphorylation sites (Figure 3C and Appendix A). Amongst FOS proteins, the C-DEST domain of zebrafish Fosl2 showed highest identity to the human protein (Appendix A). Collectively, these findings suggest that zebrafish and human FOS proteins share key functional and regulatory features.

### 2.3. Zebrafish Fos Genes Show Distinct Expression Patterns during Embryonic Development

Wholemount in situ hybridisation (WISH) analysis was used to establish the pattern of expression of each zebrafish Fos isoform during early embryonic development. Maternal *fosaa* transcripts were detected in 1-cell embryos (0 hpf), and by 12 hpf (6-somite stage), zygotic expression was observed in the brain (forebrain, midbrain, and hindbrain), presomitic mesoderm, somites, and tailbud (Figure 4A), and at 24 hpf, in the brain (forebrain, midbrain, and hindbrain), eye, migratory neural crest cells, melanoblasts, and notochord (Figure 5A,A’). From 48 hpf to 4 dpf, no specific *fosaa* expression was observed compared to the sense controls (Figure 5B–C’ and Appendix A). These results suggest a role for *fosaa* during early organ development (brain, eye) and formation of diverse cell lineages including melanocytes, smooth muscle, craniofacial bone and cartilage, cranial neurons, and glia. The pattern of *fosab* expression was similar to *fosaa* in early embryos (Figure 4B) but persisted until 4 dpf in multiple regions of the embryo (Figure 5D–F’). These included the brain, eyes, migrating neural crest cells, melanoblasts, epidermis, liver, and pancreas at 24 hpf (Figure 5D,D’and Appendix A), and the brain, heart, gut, migratory neural crest cells, lateral line neuromasts, nephron (proximal straight tubule of nephron and distal early of nephron), cloaca, somites, and tailbud at 48 hpf (Figure 5E,E’ and Appendix A), while at 4 dpf, it was expressed in the mouth, otic vesicle, peridermis, epidermis, jaws, pharyngeal arch, and caudal fin (Figure 5F,F’ and Appendix A). These observations suggest a role for *fosab* during the development of multiple organs (eye, brain, heart, liver, and pancreas) and cell lineages (e.g., melanocytes and epidermal cells).

Zebrafish *fosb* was expressed from 8 hpf (75% epiboly stage) (Figure 4C), with specific expression observed in the hatching gland and eye at 24 hpf (Figure 5G,G’). At 48 hpf, *fosb* expression was present in the brain (midbrain, pallium), heart, hatching gland, and tailbud (Figure 5H,H’ and Appendix A), while at 4 dpf, expression was also observed in the otic vesicle, pharyngeal arches, epidermis, and caudal fin, including the fin rays (Figure 5I,I’ and Appendix A).

Zebrafish *fosl1a* transcripts were not detected at early time points, but were observed in the forebrain, pancreas, and tailbud at 24 hpf (Figure 5J,J’) and in the eye, brain, peridermis, and pectoral fin bud at 48 hpf (Figure 5K,K’ and Appendix A), whereas no specific *fosl1a* expression was detected at 4 dpf (Figure 5L,L’). Compared to *fosl1a*, *fosl1b* exhibits very limited expression, which was maternally deposited at the one cell stage (0 hpf) (Figure 4E). At 24 hpf, *fosl1b* expression was localised to the hatching gland (Figure 5M,M’), while at 48 hpf, it was expressed in the hindbrain and peridermis (Figure 5N,N’). Thus, *fosl1b* may play specific roles in the development of the brain, epidermis, and hatching gland.

Zebrafish *fosl2* showed the highest early embryonic expression being detected at 12 hpf in the eyes, somites, presomitic mesoderm, mid and hindbrain, and tailbud (Figure 4F). At 24 hpf, *fosl2* was expressed in the brain (forebrain, midbrain, hindbrain, and midbrain–hindbrain barrier), eye, otic vesicle, somites, and tailbud (Figure 5P,P’ and Appendix A). Additional staining was observed in the heart, hatching gland, and posterior notochord at 48 hpf (Figure 5Q,Q’ and Appendix A), and in the epidermis, jaws, pharyngeal arch, cloaca, and fin rays at 4 dpf (Figure 5R,R’ and Appendix A).

### 2.4. Expression of FOS Genes in Adult Zebrafish Tissues

To determine the levels of *fosaa, fosab, fosb, fosl1a, fosl1b,* and *fosl2* transcripts in adult male and female zebrafish tissues, we performed qRT-PCR analysis (Figure 6) using optimised primers (Appendix A). In male zebrafish, all *fos* genes displayed weak expression in the heart and liver (Figure 6A–F). All *fos* genes showed high expression in the brain, particularly *fosb, fosl1b, and fosl2*. All *fos* genes were expressed in the spleen, where *fosb* and *fosl1b* levels were highest. In the intestine, *fosab, fosb,* and *fosl1b* levels were high, whereas *fosaa, fosl1a,* and *fosl2* showed modest expression (Figure 6). Low to moderate level of *fos* gene transcripts were observed for other tissues examined (eyes, kidney, skin, gills, and testis) (Figure 6).

Overall, the expression of *fosaa, fosab, fosb, fosl1a, fosl1b,* and *fosl2* was similar in adult zebrafish female tissues, but with modest variations compared to males (Figure 6A–F). All *fos* genes were weakly expressed in the liver of females, whereas *fosl1a* and *fosl1b* showed highest expression in the brain (Figure 6D,E). *fosb* and *fosl1b* showed moderate expression in the spleen, skin, and gills (Figure 6C,E). Amongst *fos* genes, *fosl1b* shows highest expression in the zebrafish adult female intestine (Figure 5E), followed by *fosl2* and *fosl1a,* which shows moderate expression (Figure 6D,F). Moderate to low *fos* expression levels were observed in all other female tissues, including eyes, kidney, gills, spleen, and oocytes (Figure 6A–F).

### 2.5. Fos Genes Are Induced in BRAF-Driven Melanoma in Zebrafish

The ERK MAPK pathway is a key signalling pathway regulating the transcription of human *Fos* genes [1]. To determine if this pathway also regulates expression of zebrafish *Fos* genes, we expressed the human BRAF^V600E^ to drive ERK MAPK activation in zebrafish melanocytes using the MiniCoopR transgenesis system [36], which allows rapid testing of candidate modifiers of melanoma development in F0 zebrafish generation (Figure 7A,B). Gene expression analysis on zebrafish melanoma samples revealed higher transcript levels of all *fos* genes except *fosl1b,* with *fosb* displaying the highest relative expression (Figure 7C). These findings indicate functional conservation of *fos* gene regulation by the ERK MAPK pathway in zebrafish. 

## 3. Discussion

Despite their well-documented roles as key regulators of cell fate in a multitude of normal and pathological contexts, several important aspects of FOS protein biology remain poorly understood, particularly in vivo, such as how specific FOS heterodimers regulate transcription and how their activities are regulated. Though possible, such studies are challenging to undertake in mice, the main in vivo model that has been used to study FOS protein biology. Our findings highlight the potential of zebrafish as a vertebrate model for such studies.

The six zebrafish FOS proteins share 40–65% sequence identity and 56–74% similarity to their respective human counterparts. As expected, sequence conservation was highest in the respective bZIP domains of zebrafish and human FOS proteins. However, we also found that zebrafish Fos proteins retained significant sequence conservation in other functional domains previously identified in human FOS isoforms [1], including the N- and C-terminal transcription activation domains of c-FOS and FOSB, and the C-terminal DEST that controls protein stability. The exception was Fosl1b, which lacked a C-terminal region, including the DEST domain. As FOS proteins are stabilised upon deletion or ERK pathway-dependent phosphorylation of this domain [15,17], it is likely that Fosl1b is inherently more stable than the other zebrafish FOS orthologs and that its expression is uncoupled from ERK pathway regulation. Such divergence has often been observed in paralogs arising from gene duplication events [37] and provides an evolutionary mechanism for creating unique protein functionalities.

The major mechanism of FOS protein regulation is through post-translational modification, primarily phosphorylation. We found that phosphorylation sites previously identified in human FOS proteins [1] are also conserved in zebrafish. Thus, the functionality of each FOS isoform is likely to be regulated via similar mechanisms in both species. Additionally, the conservation of regulatory elements, including phosphorylation sites, suggests that their functional impacts could be characterised in vivo using targeted genome editing approaches in zebrafish.

One of the least understood aspects of FOS biology is their roles during early embryonic development. Compared to mice, such investigations are much easier to undertake in zebrafish, as they have transparent embryos, develop externally, and are readily amenable to genetic manipulation. Our data suggest that zebrafish *fos* genes play distinct and temporally restricted roles in specific cell lineages and tissues during early development. For example, whereas sustained *fosab* and *fosl2* expression was evident over 4 days of development in multiple lineages, *fosaa*, *fosl1a*, and *fosl1b* showed more transient expression. The pattern of fosab expression suggests it may play a role during early development of embryonic skin, liver, and pancreas. In addition to *fosab*, *fosl1a* also may play roles in pancreatic development. Interestingly, all *fos* genes except *fosl1b* showed transient expression in the embryonic eyes and brain, indicating potential roles for fos genes in development of these tissues. Expression of *fosab*, *fosb*, and *fosl2* in the jaws and pharyngeal arches suggests these genes may participate in bone and cartilage development, as has been previously reported in mice [27,38]. In addition to their distinct expression patterns in the early embryo, we found that *fos* genes were expressed in a variety of adult zebrafish tissues. Interestingly, *fos* gene expression appears to differ between some male and female tissues, most notably the spleen. This finding suggests a potential role for *fos* genes in regulating sexual dimorphism in the immune system, a possibility that warrants further investigation. Of note expression of *fos* has previously been shown to be sexually dimorphic in the zebrafish brain [39].

The data from our WISH analysis highlight the potential of zebrafish as an alternative in vivo model to mice for investigating the roles of Fos proteins in vertebrates. An important goal of future studies will be to use the powerful toolkit available for genetic manipulation in zebrafish to dissect the role of specific *fos* genes in the embryo. To date, few such analyses have been undertaken with the exception of the heart, where *fosl2* was reported to potentiate the rate of myocardial differentiation from the zebrafish second heart field [40] and where cardiomyocyte-specific expression of a dominant negative AP-1 protein leads to defects in cardiomyocyte proliferation following injury [41]. Consistent with these observations, we also noted expression of *fosl2*, as well as *fosab, fosb,* and *fosl1a,* in the embryonic heart.

In mammalian cells, Fos genes are classified as inducible immediate early genes transiently induced by extracellular signals such as growth factors, hormones, and cytokines [42]. Though limited, available evidence indicates that they are also likely to be inducible genes in zebrafish. For example, expression of zebrafish c-Fos orthologs has been shown to be induced in the dorsal telencephalon by cannabinoids [43]. *Fosl1a* was induced during skeletal muscle regeneration [44] while *fosab*, *fosl1a,* and *fosl2* were induced during heart regeneration [45]. The major pathway regulating transcription of *Fos* genes in mammalian cells is the ERK MAPK pathway, whose activation in response to IGF and FGF signalling has been shown to control key developmental events in zebrafish embryos, including somite boundary formation [46], dorsoventral patterning [47], axial patterning [48], and the development of the subpallial telencephalon [49]. Interestingly, we found that *fosaa, fosab, fosl1b,* and *fosl2* were expressed in the presomitic mesoderm, where ERK signalling has been shown to be critical for somitogenesis in both zebrafish and chick embryos [50]. In addition to developmental contexts, ERK signalling has also been reported to induce *FOS* gene expression downstream of oncogenic lesions, such as activating mutations in the BRAF gene, which frequently occur in in human melanomas [51,52,53,54]. Consistent with these observations, we show that *fos* gene expression was also induced in zebrafish BRAF mutant melanomas, indicating functional conservation of key pathways regulating expression of these genes in humans and zebrafish. Thus, zebrafish may provide a useful model to dissect the role of specific *Fos* genes during development and progression of melanoma and other oncogene-driven cancers.

## 4. Materials and Methods

### 4.1. Identification of Zebrafish FOS Family Orthologues

Zebrafish orthologues of human *FOS* genes were identified by bioinformatics analysis using the National Centre for Biotechnology Information (https://www.ncbi.nlm.nih.gov/, accessed on 9 May 2022) and the Zebrafish Information Network (https://zfin.org/, accessed on 9 May 2022). The FASTA sequences of human FOS mRNAs were obtained from the NCBI database and used for Basic Local Alignment Search Tool (BLAST) (https://blast.ncbi.nlm.nih.gov/Blast.cgi, accessed on 9 May 2022) analysis to identify the zebrafish orthologues which were further crosschecked with the ZFIN database and confirmed by sanger sequencing.

### 4.2. Multiple Sequence Alignment (MSA) and Phylogenetic Comparison of FOS Family Orthologues in Human and Zebrafish

Sequence comparisons of human and zebrafish FOS protein orthologues were determined using pBLAST. Multiple sequence alignments (MSA) were performed using Clustal X version 2.1 software and bootstrapped phylogenetic trees of 1000 replicates generated using the neighbor-joining algorithm [55], then visualised in TreeView [56]. The genomic arrangement of human and zebrafish *FOS* gene loci, including syntenic genes, was determined using NCBI Map Viewer. Information on regulatory phosphorylation sites in human FOS proteins was obtained from the Phosphosite database (www.phosphosite.org, accessed on 9 May 2022). Synteny analysis for zebrafish and human *FOSL1* and *FOSL2* genes was performed using the Genomicus database [57].

### 4.3. Zebrafish Maintenance and Embryo Collection

Zebrafish were maintained under standard husbandry practices [39], following the national guidelines for animal use and care, with approval from the Deakin University Animal Welfare Committee. Embryos were collected manually and nurtured in a petri dish containing E3 media and incubated at 28.5 °C, with transparency maintained by adding 0.003% (*w*/*v*) 1-phenyl-2-thio-urea (PTU), a pigment inhibitor into the E3 media from 9 h post fertilization (hpf).

### 4.4. Quantitative Real-Time Polymerase Chain Reaction (qRT-PCR)

Adult zebrafish were euthanised with benzocaine and their tissues were isolated by dissection. Total RNA was extracted using RNeasy Mini Kit (Qiagen) according to the manufacturer’s guidelines. qRT-PCR was performed using the iTaq™ Universal SYBRGreen Supermix (Biorad, South Granville, Australia) according to the manufacturer’s guidelines with the following primers: *fosaa* (5′-AACATCAAGAAGCGAGGCGT, 5′-CGGAGACTCGCCCTGTGC), *fosab* (5′-CGATACACTGCAAGCTGAAAC, 5′-CGGCGAGGATGAACTCTAAC), *fosb* (5′-GTCAAAGTGGCACGGGCT, 5′-GGTCAGCGTTTCATCTCGTG), *fosl1a* (5′-CACAACCCAACAACAACAGAAG, 5′-GGAGGGCTGAGGAGGATTC), *fosl1b* (5′-TCAAACCCGCAAGTCACCTC, 5′-ATCTATGCTGGTTGTGAATGAC), *fosl2* (5′-GACACTGGTCGTCTGGGAAT, 5′- TACTTCTGGTAACTGGAGGCG), and *actb* (5′-TGGCATCACACCTTCTAC, 5′-AGACCATCACCAGAGTCC). Standard and unknown samples were assayed in triplicate using the following thermocycle profile conditions: initial incubation at 95 °C for 5 s, 58 °C for 10 s, and then 72 °C for 20 s. Data was normalized to *actb* and the relative fold changes in adult male and female zebrafish tissue gene expression were determined by comparison with the expression in the male eyes (tissue with moderate overall fos gene expression) by using the ddct method [58].

### 4.5. Whole-Mount In Situ Hybridization (WISH)

Total RNA was isolated from zebrafish embryos using TRIsure (Bioline) according to the manufacturer’s instructions and cDNA synthesis was performed using an iScript cDNA synthesis kit (Promega). RT-PCR was performed to amplify cDNA by using the Go Taq Green Master Mix (Promega, Madison, USA) using the following primers: *fosaa* (5′-GGAAGAGCAAGAGCGAGCA, 5′-CTTGTAGAGCGTCTCCCAGTC), *fosab* (5′-TGACAGGATGATGTTTACCAGC, 5′-CGGCTCCAGGTCAGTGTTAG), *fosb* (5′-ACAGTAGGTGTTTCGGCTTCTC, 5′-GGTGGTGGGATGAGTAGGC), *fosl1a* (5′-AAACGCCAGCAGAGAGTGTC, 5′-GGTAAATGGAGTCAGGGATGG), *fosl1b* (5′-ACACCAACCAGCCACGAAAC, 5′-CGGGAATCATAATACGACTCTC), and *fosl2* (5′-TGTCGGAACCGCAGAAGAG, 5′-ATCTCTCCTCTGGCTTTACCTTC). PCR reactions were performed under the following conditions for 35 cycles: 2 min at 95 °C, 30 s at 95 °C, 30 s at 55 °C, 1.5 min s at 72 °C, and 10 min at 72 °C. RT-PCR amplified cDNA products of *fosaa, fosab, fosb, fosl1a, fosl1b,* and *fosl2* were isolated, purified and cloned into pGEM-T Easy vectors. Purified plasmids with the gene insert were linearised, followed by transcription using either T7 or SP6 polymerase and DIG RNA labelling mix (Roche) to generate DIG-labelled anti-sense and sense probes, which was followed by their purification using Probe-Quant G-50 micro columns (Cytiva). At different time points during development, embryos were collected, dechorionated, and anesthetized with 0.4 mg/mL benzocaine before fixing with 4% *w/v* paraformaldehyde (PFA) at 4 °C. WISH was performed using digoxygenin (DIG)-labelled RNA probes as described in [38] and the images were obtained using an Olympus MVX10 monozoom microscope with a 1 × MVXPlan Apochromat lens (NA = 0.25) with an Olympus DP74 camera. Specific expression of each probe was confirmed by comparison to their corresponding sense controls.

### 4.6. Melanoma Model

Zebrafish melanoma formation was studied as described previously [36]. Briefly, 25 pg of a MiniCoopR vector expressing BRAF^V600E^ (MiniCoopR mitfa:BRAF^V600E^) was microinjected along with 25 pg of Tol2 transposase mRNA into one-cell stage *tp53* knockout embryos. The animals were monitored weekly for the presence of visible tumours. The tumours were dissected and RNA extracted for subsequent qRT-PCR analysis to determine the expression of zebrafish *fos* genes.

## Figures and Tables

**Figure 1 ijms-23-10098-f001:**
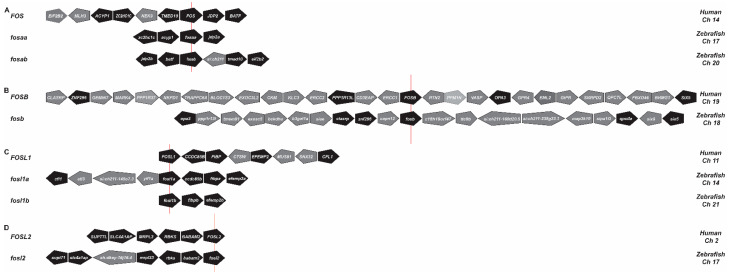
Synteny analysis of *FOS* genes. Human and zebrafish *FOS* (**A**), *FOSB* (**B**), *FOSL1* (**C**), and *FOSL2* (**D**) gene loci, indicating adjacent genes in their respective orientations. Zebrafish genes that share conserved synteny between human neighbouring genes are shown in black and non-syntenic genes in grey. The red line indicates the reference gene. Ch, chromosome.

**Figure 2 ijms-23-10098-f002:**
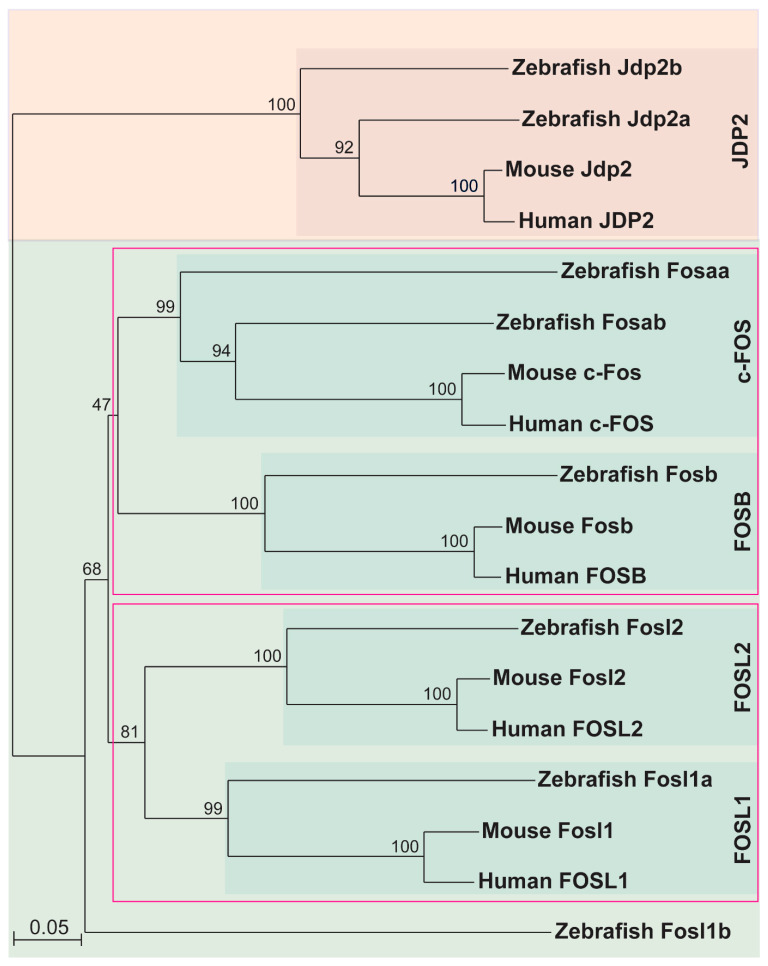
Phylogeny of mammalian and zebrafish FOS proteins. Phylogenetic analysis of human, mouse, and zebrafish FOS proteins were performed using the neighbour-joining algorithm and visualised with TreeView, with bootstrap values represented as a percentage of 1000 replicates and the relative evolutionary distance represented at the bottom left corner. JDP2 proteins were used as a closely related outgroup. Clades of related proteins are indicated within pink coloured boxes.

**Figure 3 ijms-23-10098-f003:**
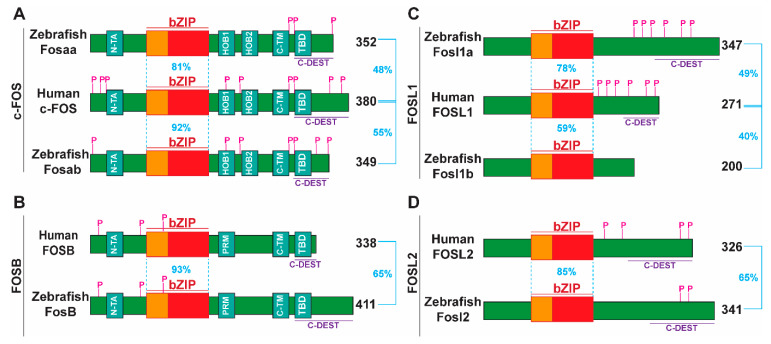
Schematic representation of human and zebrafish FOS proteins. Depicted are human and zebrafish sequences related to c-FOS (**A**), FOSB (**B**), FOSL1 (**C**), and FOSL2 (**D**). Functional regions indicated are the basic leucine zipper (bZIP) including the basic region for DNA interaction (orange box) and leucine zipper for dimerization (red box), N-terminal transactivation domain (N-TA), homology box one and two (HOB1, HOB2), proline-rich motif (PRM), C-terminal transactivation motif (C-TM), transactivation domain (TBD), and destabiliser region (C-DEST). Previously characterised phosphorylation (P) sites curated in the Phosphosite database (phosphosite.org (accessed on 24 July 2022)) are indicated. The percentage identity within the bZIP domain and overall protein are indicated by blue dotted or solid lines and text, with the protein length indicated on the left in black text.

**Figure 4 ijms-23-10098-f004:**
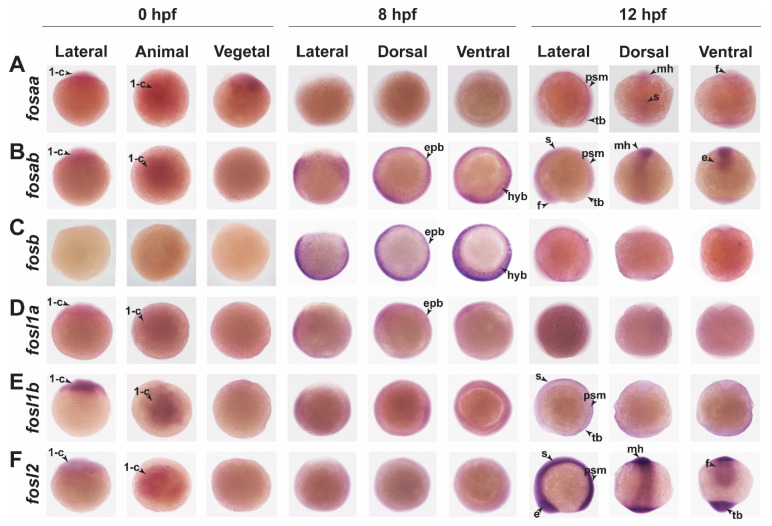
Spatiotemporal expression of *fos* genes during early zebrafish embryonic development. Lateral, dorsal, and frontal views of 0 hpf, 8 hpf, and 12 hpf zebrafish embryos analysed by WISH showing expression for *fosaa* (**A**)*, fosab* (**B**)*, fosb* (**C**)*, fosl1a* (**D**)*, fosl1b* (**E**), and *fosl2* (**F**). 1-c, one cell; e, eye; epb, epiblast; f, forebrain; hyb, hypoblast; mh, mid and hindbrain; psm, presomitic mesoderm; s, somite; tb, tailbud. Black arrows within each panel point to the specific expression indicated by the abbreviation.

**Figure 5 ijms-23-10098-f005:**
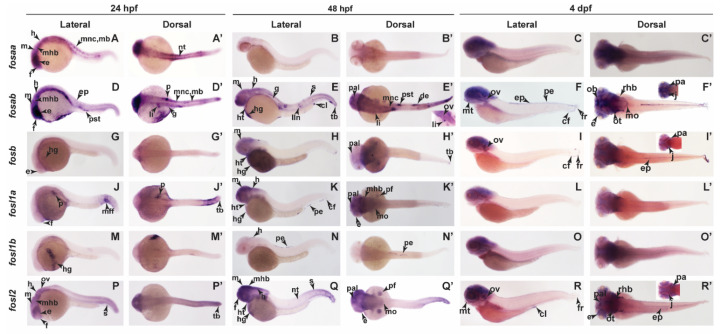
Spatiotemporal expression of *fos* genes in later stages of zebrafish embryonic development. Lateral and dorsal views of zebrafish embryos analysed by WISH showing expression for *fosaa* at 24 hpf (**A**,**A’**)*,* 48 hpf (**B**,**B’**), and 4 dpf (**C**,**C’**); *fosab* at 24 hpf (**D**,**D’**)*,* 48 hpf (**E**,**E’**), and 4 dpf (**F**,**F’**)*; fosb* at 24 hpf (**G**,**G’**)*,* 48 hpf (**H**,**H’**), and 4 dpf (**I**,**I’**); *fosl1a* at 24 hpf (**J**,**J’**)*,* 48 hpf (**K**,**K’**), and 4 dpf (**L**,**L’**)*; fosl1b* at 24 hpf (**M**,**M’**)*,* 48 hpf (**N**,**N’**), and 4 dpf (**O**,**O’**) and *fosl2* at 24 hpf (**P**,**P’**)*,* 48 hpf (**Q**,**Q’**), and 4 dpf (**R**,**R’**). cf, caudal fin; cl, cloaca; de, distal early of nephron; e, eye; ep, epidermis; f, forebrain; fr, fin ray; g, gut; h, hindbrain; hg, hatching gland; ht, heart; j, jaw; li, liver; lln, lateral line neuromasts; m, midbrain; mb, melanoblast; mff, median fin fold; mhb, midbrain hind brain barrier; mnc, migratory neural crest cell; mo, medulla obolongata; mt, mouth; nt, notochord; ob, olfactory bulb; ot, optic tectum; ov, otic vesicle; p, pancreas; pa, pharyngeal arches; pal, pallium; pe, peridermis; pf, pectoral fin; pnc, posterior notochord; pst, proximal straight tubule of nephron; rhb, rostral hindbrain; s, somite; tb, tailbud. Black arrows within each panel point to the specific expression indicated by abbreviations.

**Figure 6 ijms-23-10098-f006:**
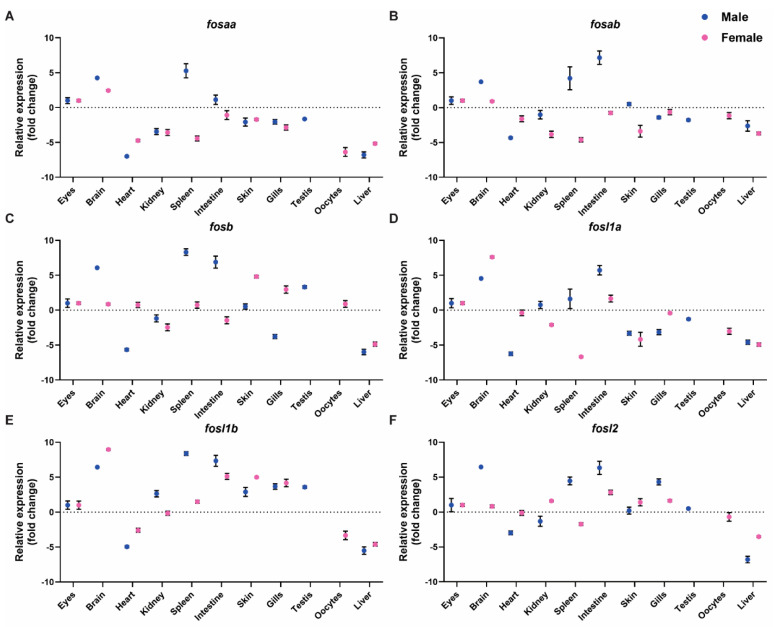
Relative expression of *fos* genes in male and female zebrafish adult tissues. mRNA expression levels of *fosaa* (**A**)*, fosab* (**B**)*, fosb* (**C**)*, fosl1a* (**D**)*, fosl1b* (**E**), and *fosl2* (**F**) in zebrafish adult male tissues eyes, brain, heart, kidney, spleen, intestine, skin, gills, testis, and liver were determined by qRT-PCR. The relative mRNA levels were normalised to *actb* and ddct values determined by comparison with expression in the male eyes, the tissue with moderate expression. The black error bars indicate the standard error of mean for four biological replicates and the dotted line indicates the median in a tissue showing moderate expression.

**Figure 7 ijms-23-10098-f007:**
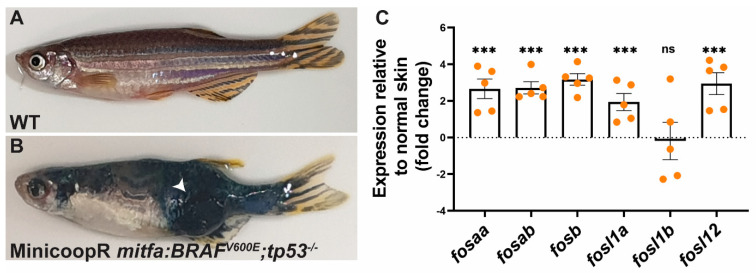
Relative expression of *fos* genes in human BRAF^V600E^ oncogene driven melanoma in zebrafish. mRNA expression levels of zebrafish *fos* genes were determined in normal wildtype (WT) skin (**A**) and melanoma tumour from BRAF^V600E^; *tp53*^−/−^ zebrafish line (**B**), with the graph showing the expression relative to the normal skin (fold change) (**C**). The relative mRNA levels were normalised to *actb* and ddct values determined by comparison with expression in the normal wildtype skin. Error bars indicate the standard error of mean for five biological replicates with statistical significance indicated (*** *p* < 0.001, ns not significant). The white arrow indicates the tumour.

## Data Availability

All data generated or analyzed during this study are included in this published article (and its Appendix A).

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
