# Peer review of "Phylogenetic and Expression Analysis of Fos Transcription Factors in Zebrafish"

_ijms, 2022, doi:10.3390/ijms231710098_

Round 1

Reviewer 1 Report (Previous Reviewer 1)

It lacks some mechanistic insights and better quality images for WISH, however the molecular part is reasonable and in line with the special issue.

Author Response

28 August 2022

RE: Manuscript ijms-1901036

Reviewer 1

 It lacks some mechanistic insights and better quality images for WISH, however the molecular part is reasonable and in line with the special issue.

We have provided higher resolution images of the WISH data in the revised manuscript.

Reviewer 2 Report (Previous Reviewer 3)

The re-submitted article by Kubra et al has been improved by additional experiment showing significance of BRAF-ERK pathway (Figure 7) in activation fos orthologs in zebrafish. As previously some reviewers pointed out the article was not showing mechanism of FOS genes regulation. Currently, authors drive conclusion that indeed BRAF-Erk pathway might be responsible for regulation of those genes as known in human. Despite it is a very basic finding I believe that it is of great importance to show that there is a functional alternative model for other organisms. We can argue that from this work it is not known whether and/or how expression (downregulation or overexpression) of particular Fos members is affecting organs/cell function. Other reviewer also tried to obtain answer to sex differences in FOS gene expression, which I guess was not explained in corrected version. As for mechanistic works I believe there would be open space for additional experiments showing importance of particular phosphorylation sites, role of individual Fos members and precise downstream molecular pathways. I guess that authors did not pay too much atention to discuss e.g. why fos1lb gene was not affected by BRAF mutant. However, in general as basic work, showing the new model this article is sufficient and I would opt for its publication in IJMS.

Author Response

28 August 2022

RE: Manuscript ijms-1901036

Reviewer 2

 The re-submitted article by Kubra et al has been improved by additional experiment showing significance of BRAF-ERK pathway (Figure 7) in activation fos orthologs in zebrafish. As previously some reviewers pointed out the article was not showing mechanism of FOS genes regulation. Currently, authors drive conclusion that indeed BRAF-Erk pathway might be responsible for regulation of those genes as known in human. Despite it is a very basic finding I believe that it is of great importance to show that there is a functional alternative model for other organisms.

(1) We can argue that from this work it is not known whether and/or how expression (downregulation or overexpression) of particular Fos members is affecting organs/cell function.

Having only demonstrated sequence conservation and expression patterns of Fos isoforms in this manuscript, we only now have the rationale to commence the type of studies suggested by the reviewer. However, as indicated in our previous response to the reviewer, these experiments will take some years to complete and are thus beyond the scope of the present study.

(2) Other reviewer also tried to obtain answer to sex differences in FOS gene expression, which I guess was not explained in corrected version.

We have now added a short discussion relating to this data in the Discussion section.

(3) As for mechanistic works I believe there would be open space for additional experiments showing importance of particular phosphorylation sites, role of individual Fos members and precise downstream molecular pathways.

We agree. However, to address the potential functional significance of a particular phosphorylation in a specific zebrafish Fos isoform requires us to first define the functions of that isoform in vivo, then use genome editing to alter specific residues to see if they alter the function that protein. As indicated in our previous response to the reviewer, such experiments will take several years and are clearly beyond the scope of the present study.

(4) I guess that authors did not pay too much attention to discuss eg. why fos1lb gene was not affected by BRAF mutant.

There could be a range of reasons for this observation. We felt that discussing these without any experimental evidence or relevant supporting literature would be too speculative and detract from other, more relevant points worthy of discussion.

However, in general as basic work, showing the new model this article is sufficient and I would opt for its publication in IJMS.

Reviewer 3 Report (New Reviewer)

The manuscript "Phylogenetic and Expression Analysis of Fos Transcription Factors in Zebrafish" by Kubra and colleagues describes zebrafish orthologs of the human Fos genes and their expression patterns. I only have a few minor points:

1. In Figure 1, use of overlapping color palettes to indicate presumably distance (human) and similarity (zebrafish) is unnecessarily confusing. I would recommend using gray for nonsyntenic human and zebrafish genes (which are curiously not named or shown - what do the lightly grey pentagons on the zebrafish chromosomes mean?). Also, there must be a better way to represent similarity in color scale. Now the white slc4a1ap really stands out, but it is about average in terms of conservation I presume.

2. I believe the scale used in Figure 6 is rather unusual. What does 0 fold change mean? A log2 scale would be much better.

3. I find sexual dimorphism in the expression levels of many genes in the spleen and the intestine quite remarkable. If it is consistent with data from other model systems relevant literature should be cited. Regardless, it seems worth discussion.

Author Response

28 August 2022

RE: Manuscript ijms-1901036

Reviewer 3

(1) In Figure 1, use of overlapping color palettes to indicate presumably distance (human) and similarity (zebrafish) is unnecessarily confusing. I would recommend using gray for non syntenic human and zebrafish genes (which are curiously not named or shown - what do the lightly grey pentagons on the zebrafish chromosomes mean?). Also, there must be a better way to represent similarity in color scale. Now the white slc4a1ap really stands out, but it is about average in terms of conservation I presume.

We have now modified the figure as suggested by the reviewer.

(2) I believe the scale used in Figure 6 is rather unusual. What does 0 fold change mean? A log2scale would be much better.

Figure 6 represents ddct values that were calculated by comparing them to a moderately expressed tissue (eyes). A value of 0-fold change thus indicates that a Fos gene is moderately expressed in a particular tissue. Expression values ranging from “0 to 10” indicate moderate to high expression and values from “0 to -10” indicate moderate to low expression. We have added additional text in the figure legend to clarify interpretation of this data by the reader. We also wish to highlight that our figure was drawn using what is considered to be a standard way to represent gene expression data as indicated in the references below, including a recent publication in IJMS. Therefore, we have chosen to not modify the scale on this figure.

Basheer, F., Bulleeraz, V., Ngo, V.Q., Liongue, C. and Ward, A.C., 2022. “In vivo impact of JAK3 A573V mutation revealed using zebrafish” Cellular and Molecular Life Sciences, 79(6), pp.1-13.

Sertori, R., Jones, R., Basheer, F., Rivera, L., Dawson, S., Loke, S., Heidary, S., Dhillon, A., Liongue, C. and Ward, A.C., “Generation and characterization of a zebrafish IL-2Rγc SCID model”, International Journal of Molecular Sciences, 23:4 (2022), p.2385.

(3) I find sexual dimorphism in the expression levels of many genes in the spleen and the intestine quite remarkable. If it is consistent with data from other model systems relevant literature should be cited. Regardless, it seems worth discussion.

We have now added a short discussion relating to this data in the Discussion section.

This manuscript is a resubmission of an earlier submission. The following is a list of the peer review reports and author responses from that submission.

Round 1

Reviewer 1 Report

In this manuscript by Kubra et al., the authors identify and characterize the different genes of the FOS family of proteins in zebrafish. They also analyze their patterns of expression during development by WISH and in adults using qPCR. 

This work is interesting as it unravels for the first time the different fos genes in zebrafish and analyze the spatial and temporal expression of their corresponding mRNAs.

The analysis is overall solid, appropriate controls are considered and the study is well presented.

However, to this reviewer, this paper is purely descriptive and falls short from giving any mechanistic insight into the function or signaling pathway related to any of the genes described here to be considered for publication in IJMS.

Some of the issues that could be improved for further publication:

1-     Authors might need to be careful when it comes to genes and proteins and refer to zebrafish nomenclature guidelines (ref Figure 1). Not sure if dr is the best way zebrafish species are referred to.

2-     The in situ analysis is nicely done with appropriate controls, however most of the images are of poor quality leaving some doubt regarding some of the patterns described here. 

I encourage the authors to download some better quality images (higher magnification too) especially for 24, 48 hpf and 4 dpf embryos.

Reviewer 2 Report

In this article, Kubra et al have analyzed the various orthologs for FOS transcription factors in the zebrafish through Phylogenetic and sequence analysis. Also, the authors have shown the importance of these fos orthologs using zebrafish development using in-situ hybridization and qPCR. However, the Manuscript has serious flaws and I feel that the manuscript is not suitable for publication in IJMS even if the authors modify the manuscript. This manuscript may be suitable for publication in very specialized Journals like Gene Expression and pattern etc. 

Major Concern:

1)    Images from Insitu hybridization: The quality of images is pretty low and gene expression in specific tissue like the heart is not visible.

2)    No functional Data: Authors should show the importance of these orthologs using knockdown studies. For eg. Using the morpholino. 

3)    Authors have shown the expression of Fos orthologs using qPCR in organs extracted from adult males and females. What was the basis of analyzing fos orthologs in males and females, separately?

Reviewer 3 Report

In the submitted manuscript authors tried to provide evidence for use of zebrafish as a interesting and suitable model to study the role of FOS proteins. All the parts of manuscript provide sufficient information to follow the goal of the study, which together with results and discussion creates interesting work. Of course one of the limitations is how and if differences and similarities in protein sequenc will affect their functionality. If similarity in phosphorylation sites resemble also the mechanisms found in mice or human.

There are some minor issues that I would like to point out, which maybe authors could improve. In lines 37-41 there is a nice introduction about domains that are present or not in different FOS members. however at first it is hard to follow this part as could easier with some graphical presentation. On the other hand phylogenetic and Blast comparison between human and zebrafish proteins provide nice deatils especially on Figure 3.

I have also one question about determination of expression for qRT-PCR. Is this approach proposed by authors not having some bias? The problem seems to be that if you calculate fold change to very low expressing tissue it may be much less adequate then to intermidate tissue. This should be reconsidered, and probably tissue with moderate expression chosen (such as heart but not liver) for relative presentation of data could be more appropaite. Low signal to which other tissues were compared may create some discrepancy. By curiosity I wanted to ask what were the Ct values in liver vs brain or othere tissues?

Line 144 maybe it would sound better "Followed by their purification..."

Otherwise I believe that manuscript is well presented and after some minor corrections could be published in the journal.